# A Realistic Mixture of Persistent Organic Pollutants Affects Zebrafish Development, Behavior, and Specifically Eye Formation by Inhibiting the Condensin I Complex

**DOI:** 10.3390/toxics11040357

**Published:** 2023-04-09

**Authors:** Gustavo Guerrero-Limón, Renaud Nivelle, Nguyen Bich-Ngoc, Dinh Duy-Thanh, Marc Muller

**Affiliations:** 1Laboratory for Organogenesis and Regeneration, GIGA Institute, University of Liège, 4000 Liège, Belgium; g.guerrero.limon@pm.me (G.G.-L.); rnivelle4550@gmail.com (R.N.); duythanh84bio@gmail.com (D.D.-T.); 2VNU School of Interdisciplinary Studies, Vietnam National University (VNU), Hanoi 10000, Vietnam; nbngoc.87@gmail.com

**Keywords:** SVHC, persistent organic pollutants, POP, PFOS, zebrafish, development, behavior, condensin I

## Abstract

Persistent organic pollutants (POPs) are posing major environmental and health threats due to their stability, ubiquity, and bioaccumulation. Most of the numerous studies of these compounds deal with single chemicals, although real exposures always consist of mixtures. Thus, using different tests, we screened the effects on zebrafish larvae caused by exposure to an environmentally relevant POP mixture. Our mixture consisted of 29 chemicals as found in the blood of a Scandinavian human population. Larvae exposed to this POP mix at realistic concentrations, or sub-mixtures thereof, presented growth retardation, edemas, retarded swim bladder inflation, hyperactive swimming behavior, and other striking malformations such as microphthalmia. The most deleterious compounds in the mixture belong to the per- and polyfluorinated acids class, although chlorinated and brominated compounds modulated the effects. Analyzing the changes in transcriptome caused by POP exposure, we observed an increase of insulin signaling and identified genes involved in brain and eye development, leading us to propose that the impaired function of the condensin I complex caused the observed eye defect. Our findings contribute to the understanding of POP mixtures, their consequences, and potential threats to human and animal populations, indicating that more mechanistic, monitoring, and long-term studies are imperative.

## 1. Introduction

Sixty years ago, Rachel Carson started raising awareness about persistent organic pollutants (POPs) in her book “Silent Spring”. She documented the deleterious effects caused by the indiscriminate use of DDT. Ever since, research has proven and continues to prove her point; POPs have been listed in the “Stockholm Convention on Persistent Organic Pollutants” [1]. The European Union and the United Nations Environmental Program define the persistent organic pollutants (POPs) as “chemical substances that are hard to degrade, with a tendency to bioaccumulate, transfer through the food web rather easily, transport across international boundaries, and having long half-lives” [1,2]. They have been extensively linked to adverse health effects [3,4,5,6,7]. POPs, though very relevant for modern life, are normally studied in a reductionist approach, where a single compound is targeted and tested. Nevertheless, POPs are rarely found as stand-alone compounds in nature [8,9,10]. Mixtures are the rule [11], and their effects have not been widely described yet. To understand potential threats resulting from this exposure, studies have been carried out using chemical mixtures with different approaches ranging from molecular biology to transgenerational studies, using in vitro, in vivo, and in silico techniques. Studies recently focused on a constructed mixture of POPs that was designed based on the levels found in the blood of a Scandinavian human population [12]. Using cell reporter assays, this mixture was found to antagonize the androgen receptor transactivation and nuclear translocation [13], to inhibit the transactivation activity of the aryl hydrocarbon receptor [14], and to induce cytotoxicity while it enhances nerve-growth-factor-induced neurite outgrowth in PC12 cells at high concentrations [15]. Microscopic high content analysis (HCA) revealed that some sub-mixtures affected cell number, nuclear area, and mitochondrial membrane potential in A-498 human kidney cells [16].

In this study we have chosen zebrafish larvae due to their many technical and practical advantages. To name a few, the zebrafish (*Danio rerio*) shares a non-negligible amount of genetic pool with humans (up to 80%), small size, and ease of maintenance in captivity, high fecundity and short times till adulthood and reproduction, amongst many others [17]. In this context, research using zebrafish has shed some light on the toxicological features of chemical compounds. Therefore, the aim of this research was to employ zebrafish larvae as a model organism to describe the many adverse developmental effects caused by realistic doses of the POP mixture and of specific sub-mixtures. Furthermore, we performed RNA-seq analysis on the larvae exposed to the POP mixture, aiming at elucidating the mechanism of action for specific observations.

## 2. Materials and Methods

### 2.1. Zebrafish Husbandry and Ethical Considerations

Adult wild-type zebrafish of the AB strain and the transgenic line *Tg(kdrl-mls:GFP)* [18] were obtained from breeding facilities at the GIGA-Institute, Liege, Belgium. Fish maintenance, breeding conditions, and egg production were described in detail [19,20] and are in accordance with internationally accepted standards. Animal care and all experimentation were conducted in compliance with Belgian and European laws (Authorization: LA1610002 Ethical commission protocol ULg19-2134 and Ulg19-2135).

### 2.2. Chemicals and POP Mixture

Dimethyl sulfoxide (DMSO, >99.9%, CAS number 67-68-5) was purchased from Sigma-Aldrich (Merck KGaA, Darmstadt, Germany). The stock solutions for the total POP mixture and six sub-mixtures were designed and prepared by the Norwegian University of Life Sciences, Oslo, Norway [12] as indicated in Appendix A. Briefly, the total POP mixture was designed to represent a mixture of 29 compounds at 1,000,000-fold the mean concentrations found in the blood of a Scandinavian population, while the sub-mixtures consisted of either one single class of these compounds (PFAA, Br, Cl) or of two combined (PFAA + Br, PFAA + Cl, Br + Cl) classes. Stock solutions of POP mixtures and their sub-mixtures were prepared in DMSO and stored at −20 until the day of testing. For all treatments, we used the stock solution (1,000,000×) that was further diluted on the testing day in E3 zebrafish raising media [21]. Next, the concentration of DMSO was corrected to achieve 0.1% in all cases, including the control groups.

### 2.3. Exposure Tests

Exposure tests were performed in 6 well-plates, with 25 fertilized eggs per well in 4 mL of E3 medium supplemented or not with the test compounds. For each experiment, 150 fertilized eggs were selected, 50 as controls and 100 for the specific treatment, to ensure a sufficient number of treated individuals for the tests. Each treatment was repeated at least three times in independent experiments. To keep stable chemical concentrations, we used a static-renewal approach where at least 90% of the media was refreshed every 24 h. Exposure started between 0 to 6 h post fertilization (hpf); the larvae were treated for at least 96 h. Finally, following the guidelines of the OECD Test number: 236, we tested 8 different concentrations (1×, 5×, 25×, 75×, 125×, 250×, 500×, 1000×) to estimate the median lethal concentration LC_50_.

### 2.4. Morphological Observations

A set of morphological features was recorded, including presence of edemas, inflation of the swim bladder, eye malformations, etc. Pictures of treated and untreated larvae at different stages were taken. All observations were made with a stereomicroscope Leica M165 FC (Leica Microsystems^©^, Leica, Wetzlar, Germany). Standard length was estimated in fish at 5- and 10-days post fertilization (dpf) using FIJI line tool for measurement (ImageJ2, v. 2.3.0/1.53f).

### 2.5. Mitochondrial Toxicity

Estimation of the mitochondrial integrity in the blood vessels was conducted using the transgenic zebrafish line *Tg(kdrl-mls:GFP)*, which expresses the fluorescent protein GFP fused to a peptide targeting it to the mitochondria (MLS) under the control of the endothelial cell-specific promoter sequence of the zebrafish *kdrl* gene. Heterozygous parents for the transgene were crossed, offspring carrying the transgene were selected based on fluorescence at 24 hpf and separated in control and treated groups. Then, the exposure test was carried on as described above. Fluorescence intensity was observed, and pictures were taken at 120 hpf using the epifluorescence stereomicroscope Leica M165 FC (Leica Microsystems^©^). Then, fluorescence was quantified using FIJI. Since the transgene is expressed in all blood vessels (head, heart, etc.), to avoid overestimation of the intensity, values were obtained from sectioning the body in a lateral view and using only the tail, from the opening of the anus to the caudal peduncle. Each intensity value was determined using the corrected total fluorescence (CTF) [22] and expressed as RFU (relative fluorescence units).

### 2.6. Heart Rate

Heartbeats were counted manually using an inverted Nikon Eclipse TS100 microscope and a counter for 15 s. To obtain the beats per minute (BPM), measurements were multiplied by 4. The heart rate was estimated on 96 hpf larvae that were acclimated to the lighting conditions for no less than 5 min prior to counting; the larvae were not immobilized by anesthetics or other means. Each larva was observed sequentially at least three times. Ten larvae were observed per treatment and each experiment was performed at least in triplicate.

### 2.7. Behavior

Behavioral tests were conducted on zebrafish larvae at 98~120 hpf and every test was performed between 10:00 and 13:00 to maintain a constant position in the circadian cycle. During the entire exposure period to the chemicals, special care was taken to avoid the interference of environmental factors. Exposed larvae were shielded from loud noises, changes in the temperature of the incubator (27–28 °C) and the raising media (~26° at the time of testing), changing light conditions and activities in the room by putting them in a specific enclosure harboring its own, constant dark/light regime. Prior to each behavioral test, the zebrafish larvae were inspected under a stereomicroscope to select and transfer to the testing plates only individuals devoid of any malformation that might interfere with mobility outcome (e.g., yolk sac or pericardial oedemas, spinal aberrations, aberrations in pigmentation, and/or loss of equilibrium, etc.). The larvae were placed individually in a well of a 96-well plate and observed using a ViewPoint^®^ Zebrabox system and its tracking software (ViewPoint Life Sciences, Lyon, France). The light level was set to 20% on the ViewPoint software (7.45 klux, TES 1337 light meter), while infrared light (850 nm) was used to track larval activity. We applied a light–dark cycle that lasted for a total of 1 h and consisted of 20 min of light, allowing for the larvae to acclimate to the situation and discarded from the analysis, followed by 10 min of darkness, 10 min of light, 10 min of darkness and 10 min of light. The video and tracking software were used to screen larval locomotion behavior for 10 s intervals; the distance travelled, and the time spent active were determined and, from these parameters, the mean swimming speed was also calculated by dividing the cumulated distance travelled by the total time spent active.

### 2.8. Injection of Antisense Oligonucleotide Morpholino

As previously described [23,24], one cell-stage embryos were injected with a concentration of 100 μM of MO*^p^^53^* (MO, Gene Tools Inc., Philomath, OR, USA). The morpholino was diluted in Danieau buffer and 0.5% tetramethylrhodamine dextran (Invitrogen, Merelbeke, Belgium). To assess the effects of morpholino injection, 150 individuals were microinjected in two independent experiments, followed by exposure to chemicals as described above. Sequence of the morpholino oligonucleotide:MO^*p53*^: 5′-GACCTCCTCTCCACTAAACTACGAT-3′

### 2.9. RNA Extraction

RNA was extracted from pools of 65 larvae at 5 dpf using the RNA mini extraction kit (Qiagen, Hilden, Germany). Samples were lysed in RLT+ buffer with β-mercaptoethanol (Sigma-Aldrich, St. Louis, MO, USA) and homogenized at least 10 times with a 26-gauge needle in a 1 mL syringe. An amount of 22 μL of RNAse free water was used to resuspend total RNA. RNA extract was treated with DNAseI (Qiagen, Hilden, Germany) to avoid DNA contamination. Quantity (ng/μL) and quality (260/280 and 260/230 ratios) of each extract was assessed by nanodrop spectrophotometer measurements. Poor quality (260/280 < 2; 260/230 < 2) samples were subsequently purified by lithium chloride precipitation, followed by 2 times pellet washing with 70% ethanol, and resuspended in 51 µL of RNAse-free water and stored at −80 °C. The integrity of total RNA extracts was assessed with BioAnalyzer analysis and provided RIN (RNA integrity number) scores for each sample (Agilent, Santa Clara, CA, USA).

### 2.10. RNAseq

cDNA libraries were generated from 100 to 500 ng of extracted total RNA using the Illumina Truseq mRNA stranded kit (Illumina, San Diego, CA, USA) according to the manufacturer’s instructions. cDNA libraries were then sequenced on a NovaSeq sequencing system, in 1 ×100 bp (single end). Approximatively 20–25 M reads were sequenced per sample. The sequencing reads were processed through the Nf-core rnaseq pipeline 3.0 [25] with default parameters and using the zebrafish reference genome (GRCz11) and the annotation set from Ensembl release 103 (www.ensembl.org; accessed 1 May 2020). Differential gene expression analysis was performed using DESeq2 pipeline [26]. Pathway and biological function enrichment analysis was performed using the WEB-based “Gene SeT AnaLysis Toolkit” (http://www.webgestalt.org; accessed on 10 November 2022) based on the integrated GO (Gene Ontology), KEGG (Kyoto Encyclopedia of Genes and Genomes) [27,28], Panther, and WikiPathways databases (all accessed on 10 November 2022 via http://www.webgestalt.org). An additional database was constructed using the Gene-mutant/Phenotype database from zfin (zfin.org; accessed on 6 March 2023). The cut-off values were set for the false discovery rate (FDR) to “adjusted *p* value < 0.05” and the fold change > 1.5.

### 2.11. Data and Statistical Analysis

For the estimation of the lethal concentration (LC_50_), data were transferred to R (4.0.2) [29] and the command “*dose.p*” used in the library “MASS” [30]. Morphological and fluorescence data were transferred to Prism 9.0.0 (v86) (Graphpad, San Diego, CA, USA). Each data set was tested for normality (e.g., using a visual cue (QQ plot), D’Agostino–Darling and Shapiro–Wilk tests) and equal variances (Bartlett’s test). Thus, parametric or non-parametric tests were performed, as indicated in each case in each figure.

Raw behavioral data sets consisted of tables holding the positions of each larva in each video frame (30 frames/second). This table was first trimmed to eliminate very short, oscillating, and likely artefactual movements, and then aggregated into 10-s periods for further analysis. These data were transferred to R version 4.0.2 to analyze motility during the dark and light phases. To assess behavior, we used linear mixed effect (LME) models within the “nlme” package [31]. Three dependent variables were used, either the “mean time spent active” (seconds), the “mean distance travelled” (mm), or the “mean swimming speed” (calculated as the mean distance travelled/mean time spent active) within each 10 s period, with “compound” and “time” as the categorical and continuous independent variables, and “batch” as a random effect. The “*Anova*” command within the “*car*” library [32] was used to extract the results for the main effects whereas the “*lsmeans*” command [33] within the “*emmeans*” library was used as a post-hoc test to compare groups against one another while adjusting for the means of other factors within the model [34]. Type II sum of squares was used for the model. Two kinds of analyses were performed (see also below in results): the “startle” response including the 10 s prior to change of phase (light to dark, or dark to light) with a length of 50 s and the values obtained for 560 s after the spike (the remaining time of the phase). Confidence was assigned at α = 95% and a *p*-value of ≤ 0.05 was considered as significant, *p* ≤ 0.05 (*), ≤ 0.01 (**), ≤ 0.001 (***).

## 3. Results and Discussion

### 3.1. LC_50_–Chronic Exposure to Total POP Mix Is Lethal at Relatively High Doses

In a preliminary range-finding experiment, we exposed AB zebrafish fertilized eggs to eight concentrations (1×, 5×, 25×, 75×, 125×, 250×, 500×, 1000× the mean human blood concentration) of the total POP mixture and we monitored survival at 24, 48, 72, and 96 hpf (Figure 1A) compared to untreated controls. The median lethal concentration (LC_50_) for this mixture was calculated at 386-fold the human blood concentration (386×). Consequently, in the following experiments, we limited the concentration range to 75×, 125×, and 250× the mean human blood concentration. This may seem high considering a normal population; however, we must consider that the chorion is a protective layer, which can be crossed easily by molecules with a size below 4000 Da [35]. The POPs studied here are hydrophobic and of small size; thus, they are potentially able to cross the chorion and to exert their effects right after adding the solutions to the media. Previous experiments revealed that only about 10% of, e.g., PFOS could be found in zebrafish larvae exposed (continuously, i.e., without medium change) for 96 hrs to the compound [36], while between 0 and 16% of the nominal amounts were found in larvae exposed to the POP mixture [37]. Given their high bioconcentration values (BCF factor used to estimate the potential to bioaccumulate) and the persistent nature of these chemicals [38,39], the harmful concentrations used here may be, eventually, reached in individuals that are constantly exposed, exerting their effects in later stages in life while continuing to accumulate through the many pathways of exposure [40,41,42]. Thus, we can assume that, in our experiments, larvae are exposed to concentrations of the POP mix that may be reached in exposed populations [43].

### 3.2. The POP Mix Significantly Reduces the Standard Length of Zebrafish Larvae

We first evaluated the effect of the POP mixture on general growth by measuring the standard length of the larvae at 5 dpf after continuous exposure to the POP125× (125× human blood concentration) mix. In parallel, we also tested equivalent concentrations of the different sub-mixtures. At 5 dpf, the average size of the larvae was significantly affected, fish treated with the total POP125× mix were significantly smaller by about 10% (from 3.4 to 3.1 mm) (Figure 1C). Among the single sub-mixes, only the PFAA mix resulted in a significantly decreased standard length, similar to but slightly less than the POP125× mix. Cl and Br mixes alone did not cause a clear effect. In line with these observations, only the binary mixtures containing PFAA (PFAA + Cl, PFAA + Br) caused a similar effect on standard length comparable to PFAA alone. Cl + Br had no effect. Thus, only those mixtures containing PFAA affected the size of zebrafish larvae at 5 dpf.

Previously, exposure to PFOA 4 ng/mL has been found to decrease the body length of 3 dpf zebrafish larvae, but not at 40 or 400 ng/mL [44]. In contrast, other PFAS (PFBA, PFHxA) did cause decrease of size at 40 and 400 ng/mL. Another study revealed that PFOS or PFOA decreased total body length at 200 or 2000 ng/mL, respectively [45]. Here, we used 217 ng/mL in the 125× POP mix for PFOA, in addition to PFOS and PFHxS, to analyze the effect on standard length at 5 dpf. Taken together, there is clear evidence that PFAS affects larval growth.

After halting the exposure (at 96 hpf) we kept the larvae to grow in normal E3 medium, free of POPs until 10 dpf. We observed a significant lethality during this period, which was not assessed in the preliminary experiment (Figure 1B). Among the surviving larvae, only those exposed to PFAA + Br were significantly smaller (Figure 1D), while all other treatments including PFAA, as well as the total mix, left no survivor. PCB and PBDE congeners were previously shown to impact survival of zebrafish larvae at concentrations around 1–5 µg/mL [46,47]; we did not observe a significant lethality induced by the Cl and Br mixtures here, suggesting that the congeners present in the POP mix are indeed less toxic.

### 3.3. Common Developmental Toxic Effects Such as Edemas and Non-Inflated Swim Bladder Were Commonly Found following Exposure to POP Mix

We also looked for other developmental defects induced by the different treatments, according to the recommendations for the Zebrafish Embryotoxicity test [48]. The most striking features observed were the presence of edemas and non-inflated swim bladder. Even at the lowest concentration of the total mix (POP 75×), a relatively large proportion of the population had edemas (~70%). Using the swim bladder as another phenotypical endpoint to assess developmental retardation, a high number of fish (close to 100%) presented developmental impairment of their swim bladders after 96 hpf compared to only 50% in the controls at this stage (Figure 2). While edemas are commonly observed in toxicity assays, they may have been caused in our experiments (whether yolk sac or pericardium) by the presence of PBDE47 [49]. The lack of inflation of the swim bladder is less common; it was previously reported as being affected by exposure to PFOA at 4.7 ng/mL [50] by interfering with thyroid hormone signaling. Other chemicals present in the mixtures may in addition contribute to disruption of thyroid hormone action [50]. Underdevelopment of the swim bladder would have important ecological consequences, impeding the normal swimming of the larvae at a crucial age.

The deleterious effects we describe here were only seen after 72 hpf. In a preliminary experiment (data not shown), we tested the capabilities of early developmental disruption of the POP mix and could not find a clear effect at stages earlier than 72 hpf, almost three full days of continuous exposure. A similar observation was made previously when testing pharmaceutical pollutants, some of which exerted their effects mainly on 72 and 96 hpf larvae [21]. We hypothesized that, though the chemicals would cross the chorion and be taken up by the embryos, the absence of some targeted molecules at these early stages would make them impervious to the POPs’ effects. That would be the case of thyroid follicles that start developing after 96 hpf [51]; hence, some PBDEs would not be exerting their effects through this pathway until a later stage.

### 3.4. Striking Eye Malformation in Fish Treated with Any of the POP Mix and Its Sub-Mixes

One outstanding feature we observed was the pear-like shaped eyes, with dents in the polar regions on the eyeball of the fish treated with the POP mix (Figure 3A–C). Compared to the control group (~7%), treatment with the POP75× mix affected about 50% of affected larvae, while POP125× affected about 70% and POP250× close to 90% of the fish (Figure 3D). When we tested the sub-mixtures at 125× the mean human blood concentration, we observed that each of the single mixes caused a slightly lower fraction of affected individuals (Figure 3E) compared to the total POP125×. However, the dual combinations PFAA + Br or PFAA + Cl reached similar levels to POP125×; addition of either Br or Cl significantly increased the incidence of this malformation relative to PFAA alone, indicating that each sub-mixture contributed to various degrees to the effect caused by the POP mix.

This malformation of the eyes was one of the most striking and unexpected effects. Previous studies have shown a link between eye malformation and certain compounds or the suppression of expression of certain genes. Two main eye malformations are described in the literature: either eyes were absent (anophthalmia), or their size was reduced (microphthalmia). The first is linked to the absence of genes such as *chokh/rx3* [52], while the second is linked to the expression of many different genes, such as *sox2* [53], *otx2* [54], *pax6a*, or *pax6b* [55]. Regarding chemical exposure, these two morphological aberrations have been described after treatments with a variety of chemicals, such as phenylthiourea [56], gold nanoparticles [57], di-butyl phthalate [58], and PCBs (Aroclor 1254) [59]. Retinal defects have been shown in workers exposed to solvents or heavy metal, and defects in photoreceptor cells were described in zebrafish exposed to PBDEs or PCBs [60]. However, to the best of our knowledge, this is the first time the pear-like shape and microphthalmia are described as a malformation caused by these kinds of mixed organic pollutants. We did not observe a correlation of this malformation with any of the other defects that we observed, indicating that a specific mechanism is involved.

One previously described zebrafish mutant, the *cap-g*^s105^ mutant, presents a reduction of retinal cell number and smaller eyes similar to what we observed here [61]. The *cap-g* gene codes for a component of the condensin I complex involved in the regulation of chromosome condensation and segregation during mitosis. The *cap-g*^s105^ mutation of this gene causes increased apoptosis in proliferating retinal stem cells, leading to a small eye phenotype, which could be partially rescued by interfering with the expression of the pro-apoptotic gene *p53*. We thus decided to test the effect of the POP125× mix on zebrafish larvae that had been previously micro-injected with antisense morpholino directed against the *p53* gene. Although injection of the MO^*p53*^ alone seems to generate some eye deformities on its own, the eye malformation induced by POP125× was significantly reduced in fish injected with MO^*p53*^ (Figure 3F), similar to what was observed for the *cap-g*^105^ zebrafish mutant [61]. Further support for this mechanism and the genes involved in this striking phenotype will be given in the transcriptome analysis section below.

### 3.5. Heart Rate Is Severely Affected after 96 h of Exposure, Especially When PFAAs Were Present

Next, we tested the effect of the total POP mix on the heart rate of the zebrafish larvae at 96 hpf. We witnessed a significant, dose-dependent increase in the heart rate upon treatment with the total mixture (Figure 4A). Testing the sub-mixtures at 125× concentration, the most pronounced effects were observed in those treatments where PFAA mix was present (Figure 4B). A weaker, but significant difference was observed as well with the Cl mix, while no significant effect was found using only the Br mix.

All binary mixtures (PFAA + Br, PFAA + Cl, Br + Cl) significantly increased the heart rate; however, the PFAA mix exerted a dominant effect while the increase caused by Br + Cl was clearly lower compared to those due to any of the treatments where PFAA was present. The highest heart rate was recorded in the fish exposed to the binary mixture PFAA + Br (197 ± 17 BPM vs. Control = 157 ± 9 BPM).

PFOS and PFOA have been shown to increase the heart rate in 72 hpf zebrafish larvae at, respectively, 500 ng/mL and 75 µg/mL [62], indicating that PFOS may be the main agent here. Chemicals such as perfluorononanoic acid (PFNA) can alter gene expression linked to cardiac development by dysregulating genes such as *amhc*, *nppa*, *nkx2.5*, *edn1* and *tgfb2* [63], but no effect on heart rate was shown. Similarly, there are 12 dioxin-like PCBs (e.g., PCB 118) that have been associated with heart conditions such as hypertension and cardiac defects [64,65]. These effects have been linked to the activation of the aryl hydrocarbon (AhR). More interestingly, even small doses of 1,2,5,6-tetrabromocyclooctane (HBCD) cause arrythmia through dysregulating the function of sarcoplasmic/endoplasmic reticulum Ca^2+^ ATPase (SERCA2a) [66]. The latter is encoded by the *atp2a2a* and *atp2a2b* genes, of which only *atp2a2a* was significantly induced after exposure to POP125× (see below). Thus, the role of Atp2a2 in inducing arrythmia is very likely, although the precise mechanism remains unclear. Finally, the cardiotoxic effects of organochlorine pesticides have been clearly described previously [67]. The mechanisms may vary but most of the compounds within the different mixes undoubtedly have the potential to alter the heart rate, leading to cardiac conditions such as arrythmia, hypertension, and other cardiac defects.

### 3.6. Fish Were Hyperactive and Responded Notably to Changes in Illumination

To test for behavioral effects, indicative of potential neurological defects, we used a standard 10-min light–dark swimming activity protocol to assess the parameters “time spent active”, “distance travelled”, and “swimming speed” in 10 s intervals. Due to the high number of multiple malformations in the larvae in POP250x, this dose was excluded from the behavioral analyses.

#### 3.6.1. Dark–Light Response

In Figure 5, we illustrate the changes in behavior observed after treatment with the total mix POP125× compared to the control, untreated larvae at 5 dpf. As expected, we observe an increase in all parameters in control larvae during the dark phase, compared to the light phase. We can also observe the initial increase in all parameters at the start of the dark phase, which decreases in time while the larvae acclimate to the new situation. This decrease is even much faster in the POP125×-treated larvae. In contrast, we observe a stronger spike in activity when switching to the light phase, indicating that the larvae do perceive the change in lighting conditions, but rapidly return to a slightly higher activity compared to the dark phase.

The effects recorded during the dark and light phases (excluding the spikes, see below) were different depending on the compounds used (Figure 6). For instance, POP125× was the only compound that increased the swimming speed (SWS) during the dark phase, whereas binary mixtures containing Cl (PFAA + Cl and Br + Cl) decreased the speed significantly compared to controls (Figure 6A). Cl alone decreased the speed significantly only when compared to PFAA, itself slightly, but not significantly, higher than control. Thus, it appears that Cl was mainly responsible for decreasing swimming speed during the dark phase. During the light phase, more striking effects on the swimming speed were seen. Overall, all treatments (except Br) caused a faster swimming speed; however, only those mixtures where PFAA was present caused a significant increase compared to the control group. Fish exposed to the POP125× mix were swimming the fastest, followed by the binary mixtures PFAA + Br and PFAA + Cl, and finally PFAA (Figure 6B). Thus, while only PFAA alone caused a significant increase in swimming speed, addition of Cl or Br in the binary mixtures further enhanced this effect.

Time spent active (TSA) was similarly significantly affected (Figure 6C,D). POP125× caused the highest increase in activity in both the dark and light phases. During the dark phase, the presence of PFAA and PFAA + Cl significantly decreased TSA values, similar to the Cl and Br + Cl groups. During the light phase, a dramatic increase in activity was observed in fish exposed to POP125×, while PFAA, PFAA + Br, and PFAA + Cl caused a significant, but weaker, increase. Br, Cl, and Br + Cl did not affect TSA relative to control, they were thus significantly different from PFAA.

The distance travelled (DT) during the dark phase was significantly higher in fish treated with POP125×, but lower in fish treated with PFAA + Cl and Br + Cl. During the light phase, a large increase in DT was seen upon POP125× treatment, while PFAA, PFAA + Br, and PFAA + Cl, caused weaker, but still significant, increases, similar to what was observed for the TSA (Figure 6E,F).

Behavior is a complex endpoint, hard to analyze, and where many variables could be playing a role and inducing changes. One of the first hypotheses we thought of to explain the altered behavior was related to compounds binding to brain aromatase or Cyp19a1b. This protein (or its isoforms) is present from 24 hpf [68] and several studies have reported changes in swimming behavior triggered by compounds such as fadrozole (a well-known aromatase inhibitor) or other endocrine disrupting chemicals like a PCB mixture (aroclor 1254), PBDE-47, or PFOA in various fish species [69,70,71,72,73]. However, involvement of classical endocrine disruptors was ruled out for environmental effects, either due to the high concentrations used (fadrozole), or to the observation that hormone antagonists did not revert the changes [73]. In our experiments, the most obvious alterations of behavior were caused by the PFAA sub-mix (with PFOA and PFOS at the highest concentrations in the mix). We also observed a significant effect of Br or Cl, also adding to the effect when used in combination with PFAA, although never reaching the extent of the full POP125× mix. Thus, we cannot discard eventual synergistic or additive effects caused by the presence of the other chemicals within the mix. The behavior altering properties of PFAAs have been described before, in various settings. At very low concentrations between 7–700 ng/L, PFOS or PFOA led to decreased activity in 5 dpf larvae when tested alone, but increased activity when both compounds were tested together [72]. Increased activity was also observed for PFOA at 400–4000 ng/mL [44] or at 300 to 2000 ng/mL considering only the dark phase [37], consistent with our findings. These authors also suggested that hyperactivity was linked to alterations in calcium signaling involving the ryanodine receptor Ryr and affecting muscular contractions. Also, sensitization of the RYR by PCBs (e.g., 28, 138 and 153) can cause a developmental neurotoxicity [74], hence affecting the photomotor response of the larvae.

#### 3.6.2. Startle Response

As mentioned above, we noticed that each time fish were recorded, we could observe a dramatic increase in their activity (spike) at the moment of drastic transition from light to dark or back to light. Thus, we decided to focus on this spike response by analyzing only the 50 s around the transition. For the light–dark transition, we observed an increased startle response for the POP125× mix, which was even higher for the PFAA sub-mix, but somehow attenuated by addition of Br in the PFAA + Br sub-mix and Cl in the PFAA + Cl sub-mix reaching significance only for swimming speed (Figure 7). Interestingly, Br alone had no effect, while Cl alone significantly increased all parameters relative to controls. In the dark–light transition, this increased response was also observed, this time higher for the POP125× mix compared to all other mixtures containing PFAA. Br and Cl alone or in combination had no effect at all compared to control, while only marginally modulating the effect of PFAA in binary mixtures.

Taken together, our results indicate that the immediate startle response, presumably corresponding to the larvae reacting to any change in environmental conditions, is amplified by the presence of POPs, especially PFAA. This is consistent with the overall higher activity, as described above; however, the Br and Cl components seem to modulate this startle response more strongly. PBDEs such as BDE-47, -99, -100, and -153 have been shown to alter behavior at concentrations as low as 2.5 µg/mL depending on the congener [46]. According to these authors, the drastic response of the larvae in lighting transition can be explained by alterations in the glutamatergic transmission and changes in electrical coupling in the presence of PBDEs or PCBs.

### 3.7. Mitochondria Responded Notably to POP Mixture

To test the effect of the POPs on mitochondria, we used the transgenic line *Tg(kdrl-mls:GFP)*, which expresses the green fluorescent protein GFP in the endothelial cells of the vasculature and directs it to the mitochondria via its Mls signal peptide [18]. Note that the expression of the endogenous *kdrl* gene is not significantly affected by POP treatment (see below and Appendix A). Exposure of these embryos to POP mixes led to a significantly increased fluorescence with POP125×, while only combined sub-mixes PFAA + Br and Br + Cl caused a significant increase (Figure 8). The higher levels of activity in mitochondria seem to be linked to OCPs and PCBs, which are explained by a POP-induced imbalance in redox, hormone homeostasis, and mitochondrial dysfunction; the mechanisms are discussed in further detail in [75,76].

### 3.8. Gene Expression Is Severely Affected by Exposure to POPs

To gain further insight into the molecular mechanisms affected by POP exposure, we compared the whole genome transcriptome of control larvae to that of larvae treated with POP75× or POP125× by RNA-Seq analysis on whole larvae at 5 dpf. The number of differentially expressed genes (DEGs at p_adjust_ < 0.05) was 172 and 2466 for, respectively, POP75× and POP125× treatments, with 1531 genes that were upregulated and 935 that were downregulated by POP125×.

Interestingly, the huge majority (169/172) of DEGs affected at the lower concentration were also, and more strongly, affected at the higher concentration (Figure 9A, see also Appendix A). As an example, the *fbxo32* (involved in muscle morphogenesis and homeostasis, and in larval behavior), *fosb* (transcription factor of the AP1 family), and *cdca7a* (regulation of hematopoietic stem cell differentiation, thymus development) genes were not significantly affected at POP75× (log(fold-change), respectively, 0.64, 0.57, and −0.65, p_adj_ > 0.05), but were strongly affected at POP125× concentration (log(fold-change) of, respectively, 2.45, 2.20, and −1.31, p_adj_ << 0.05) (Figure 9B). Another such gene is *calcoco1b,* coding for a calcium binding protein acting as a translation elongation factor.

The most highly, and most significantly, regulated genes are mostly upregulated genes. Among the most highly induced genes, we observe *fosl1a*, *fosb*, and *junba* which together form the AP1 transcription factor regulating cell proliferation, differentiation, and stress response [77]. Also, among these upregulated genes are the two paralogs *igfbp1a and igfbp1b*, coding for Igfbp proteins that interact with insulin-like growth factors (IGFs) to stabilize them and modulate their effects on growth and glucose metabolism. The *cyp24a1* gene, coding for a 1,25-dihydroxyvitamin D3 metabolizing enzyme, is significantly upregulated, possibly in relation to the growth retardation observed [78,79]. In contrast, *cyp2aa9* and *cyp2aa8*, coding for xenobiotic metabolizing enzymes [80,81,82], are among the most downregulated genes, indicating a response to inflammatory status [83]. In addition, many more genes (2297) were differentially expressed in POP125×-treated larvae relative to control, although we did not observe substantial lethality at this stage.

Pathway and gene ontology (Appendix A) analysis points to a dysregulation of the cell cycle, but also of central nervous system development, motor activity, growth, response to stress, and metabolic processes, in particular insulin signaling and glucose metabolism (Appendix A). Using the list of genes involved in insulin signaling (Appendix A) in GENEMANIA, we constructed a network of co-expressed genes encoding proteins with physical interaction (Figure 10). Strikingly, all these genes are significantly upregulated by POP125×, while also distributing in several modules.

One of these modules centers around *gys1*, indicating increased glycogen synthesis activity. Other modules involve the insulin receptor (*insrb*), the Glut1 glucose transporter (*slc2a1b*) [84], a transcriptional regulatory module centered on the serum response factor (*srfa*), or regulatory protein kinases such as the MAPkinases *map3k10*, *map3k21* and *mapk13* genes, the protein kinase C gene *prkcdb* or the serum/glucocorticoid regulated kinase 1 gene (*sgk1*). Together, these observations indicate an increase in metabolism in the POP-treated larvae.

As may be expected from exposure to environmental toxins, oxidoreductase molecular function was identified as significantly affected, with all the genes in the list interestingly downregulated. These genes code for detoxifying enzymes such as alcohol dehydrogenases (*adh5*, *adh8b*), aldehyde dehydrogenases (*aldh16a1*, *aldh9a1a.1*), or cytochromes (*cyb5r3*, *cyp2aa3*, *cyp2r1*, *cyp4v8*, *cyp51*).

Gene enrichment analysis based on the mutant descriptions at zfin (zfin.org; accessed on 6 March 2023)) revealed that mutations in a significant number of the affected genes cause defects in development of the nervous system and the eye, as well as in the mitotic cell cycle (Appendix A). Based on these results, we used the GENEMANIA database to construct networks for the genes involved in eye and brain development (Figure 11). In both networks, we notice that the genes are all downregulated and that they build a tight network of co-regulated genes. About half of the genes affecting either eye or brain development are common to both networks. When we focused on the genes whose products were shown to physically interact (Figure 11), some similarities and some differences were observed. One common module is formed by the *aurka* (aurora kinase A, a histone serine kinase), the histone deacetylase gene *hdac8*, the polo-like protein serine/threonine kinase gene *plk1*, and the *fbxo5* gene coding for a predicted ubiquitin ligase inhibitor. In the brain, this module is connected through the CDK–cyclin pair Cdk1-Ccnb1 to the gene products of *rpa1* and *smc4*, predicted to be involved, respectively, in DNA repair, replication, and chromosome organization. Two smaller modules are formed by the *birc5a*, *cdca8*, *nono*, and *sfpq* genes, all of which were shown to affect both brain and eye formation when mutated (zfin.org). In the eye, the *cdk1* gene is connected to several members of the condensin I complex, including Smc2, Smc4, Ncapg, Ncaph, and Ncapd2 (Figure 11). This observation is reminiscent of the putative involvement of the *cap-g* gene in the eye defect that we observed. Interestingly, not only mutation of the *cap-g* gene, but also morpholino knockdown of the *capd2* and *caph* genes (coding for two other components of the condensing I complex), led to reduction of retinal cell number and smaller eyes similar to what we observed here [61].

Note that the description of the *cap-g* mutant also mentions a behavioral effect, presumably based on oculomotor and optokinetic tests mainly revealing visual impairment [61]; thus, we cannot rule out that the eye defect we observed may impact the larval behavior.

Previous studies involving RNA-Seq analysis of whole larvae exposed to the POP mixture in a similar setting, albeit at lower concentrations, revealed interesting results. The PPAR pathway was shown to be affected [37], similar to our observation that the nuclear receptor genes *pparda* and *ppargc1a* were upregulated (log(fold) 0.89 and 0.64, respectively) along with those for other nuclear receptors such as *rxrab*. Conversely, our data set did not reveal calcium ion transport or signaling to be significantly affect, nor the genes involved in ryanodine receptor signaling such as *ryr1a*, *ryr1b*, *ryr2*, *myl7, actc1*, or *tnnc1*.

Another concern with environmental pollution by POPs is their effect on sexual maturation of growing embryos. While we did not perform fertility experiments in our larval development tests, data concerning the effects of sexual hormones on 2–4 dpf larvae are readily available [85,86]. Among the genes most highly upregulated by estradiol (E2) [85], only *vtg1* was moderately and non-significantly downregulated (Appendix A), other vitellogenin genes and the aromatase gene *cyp19a1b* were not affected.

In contrast, some of the most regulated genes by 17-ß-testosterone [86] were also significantly affected upon POP125× treatment, such as *insig1*, *col10a1a*, *matn1*, *tmx3a*, *gnb3a*, or *gng13b*. Altogether, these results tend to argue for the presence of an androgen antagonistic activity in the POP mix, as was previously shown in cellular assays [13].

The effect on retina and eye development is particularly interesting. The observed defect phenocopies exactly that previously found in a *capg* mutant and in morphants for *capd2* and *caph* [61], all components of the condensin I complex specifically required for retina and eye formation. Microinjection of *p53* morpholino oligonucleotides at the one–two cell stage was able to rescue the *capg* phenotype [61], but also the eye defect in larvae exposed to POP125×. Whole larvae RNA-Seq revealed that neither the *p53* gene *tp53* expression was affected, nor did the term apoptosis appear in pathway analysis, further supporting the notion that decreasing condensin I action will specifically induce *p53*-dependent apoptosis in the developing retina.

## 4. Conclusions

This study sheds some new light on the effects of a realistic mixture of POPs on vertebrate physiology, with a focus on early developmental stages. Although informing on the effect on wildlife when present in the environment, our findings are also relevant for pregnant women relative to the health of the unborn child.

As discussed above, we believe that the concentrations tested here (POP75× and POP125×) represent realistic doses that may be reached in the environment, but also in human blood due to the stability and bioaccumulation characteristics of these compounds. Furthermore, we have been using a mixture, and sub-mixtures thereof, that represent the composition of that found in a Scandinavian population. Although not directly transposable to humans, our results obtained on zebrafish larvae allow us to identify potential risks to human fetuses and can inform about the molecular mechanisms that may be involved.

The effects that we observed, and discussed above, at non-lethal concentrations of the POP-mix are growth and developmental retardation as illustrated by the retarded inflation of the swim bladder, increased heart rate, increased metabolism, and mostly increased behavioral activity. Differential expression of genes upon exposure to POP125× was indeed consistent with an increase of metabolism (insulin signaling), but further revealed impacts on musculo-skeletal development and function, brain development, and several signaling pathways. Additional studies will be required in the future to investigate which component of the POP mix is responsible for any specific effect and through which molecular mechanism it acts.

The effects observed here are mainly due to the PFAA sub-mix; this single sub-mix, alone or in combination, caused effects close to those observed with the total POP mixture. This is of high relevance at the present time, as these substances (per- and polyfluoroalkyl acids) are intensely scrutinized as the most harmful chemical group for their deleterious effects on environment and human health. Initiatives and regulations have been introduced very recently by the European Chemicals Agency [87] and the US Environmental Protection Agency [88] to investigate the origins and effects of these compounds, aiming at reducing or banning their use. Furthermore, the spotlight of public opinion has been drawn to the topic by recent movies such as “Dark waters” [89] inspired by a story in the New York Times [90] and a recent report in The Guardian [91].

Contributions of the Cl and Br sub-mixes were observed, enhancing the effect of the PFAA mix for eye malformations, behavior, mitochondrial effect, and weakly for heart rate. One interesting exception was observed for the startle response during the light to dark transition, where the PFAA sub-mix alone caused a stronger increase compared to the POP mix, which was attenuated upon addition of either the Cl or Br sub-mix in the binary mixtures.

In conclusion, our study highlights the need to study environmental pollution, not only on single compounds, but rather to consider the more realistic situation of exposure to mixtures. One striking result from our studies is that, although some of the compounds within our mixtures are described as endocrine disrupting chemicals, we did not observe significant estrogenic effects. This illustrates the fact that individual compounds in the mix may antagonize the activities of other chemicals. Our results comparing specific sub-mixtures to the total mixture further support this conclusion, with sub-mixes either enhancing or suppressing the effects of another. Finally, we also reveal a novel defect caused by POP contamination on eye development, which we propose to result from inhibition of the condensin I complex.

## Figures and Tables

**Figure 1 toxics-11-00357-f001:**
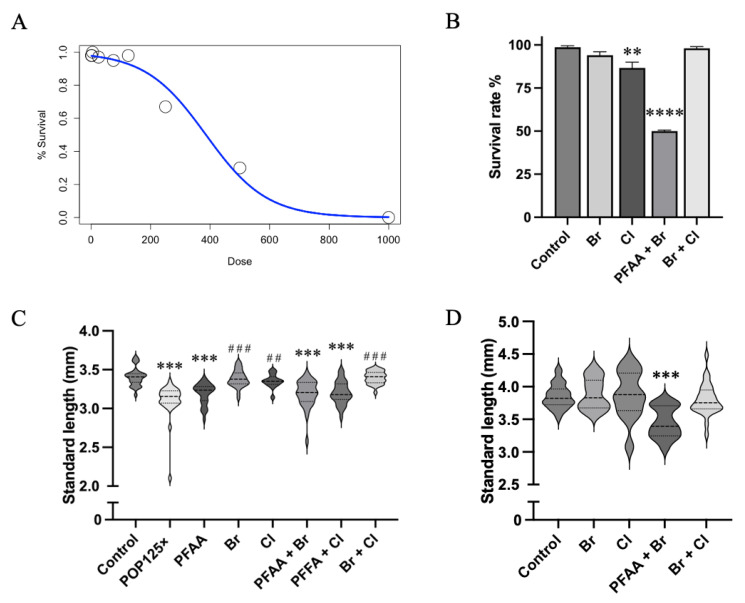
(**A**) Survival after 96 h of exposure to POP mixture. Survival decreases drastically at 386×. (**B**) Survival rate for POP mixtures and sub-mixtures at 10 dpf; ordinary one-way ANOVA and Dunnett’s multiple comparisons test. (**C**) Standard length of fish at 5 dpf. Data presented as median with higher and lower quartiles for each treatment. Asterisks (*) indicate when significant differences were found compared to control, hash sign (#) when differences were found relative to PFAA alone. (**D**) Fish measured at 10 dpf. Missing groups due to high mortality rates were not included. Kruskal–Wallis and Dunn’s multiple comparison tests, *p* ≤ 0.01 (**), *p* ≤ 0.001 (***), *p* ≤ 0.0001 (****). In short, PFAA < Total Mix < Cl = Br = Control.

**Figure 2 toxics-11-00357-f002:**
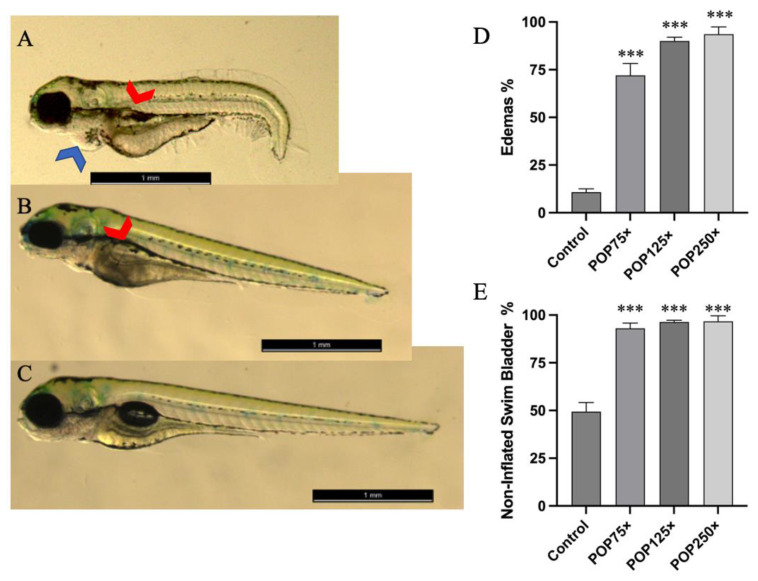
Examples of fish treated and untreated with the POP mix. Several malformations and size differences are striking. Pictures were cropped purposefully to enhance the differences found. (**A**) Fish treated with POP250x, red arrow pointing at the non-inflated swim bladder, blue arrow pointing at a pericardial edema. (**B**) Fish treated with POP125×. (**C**) Control, size bar = 1 mm. (**D**) Edema, as percentage of the population having this malformation. (**E**) Non-inflated swim bladder at 96 hpf, as percentage of population. Data presented as mean percentage of population having either malformation and standard deviation; ordinary one-way ANOVA and Tukey’s multiple comparison test, *n* = 100, *p* ≤ 0.001 (***).

**Figure 3 toxics-11-00357-f003:**
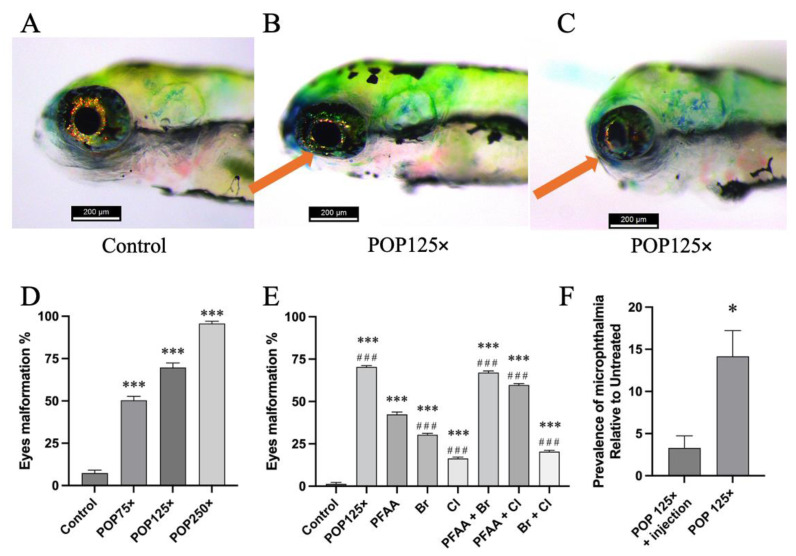
Example of eye malformation. (**A**) Untreated larva 4 dpf; (**B**) flattened eye with dents at both the upper and lower side of the eye and (**C**) eye hypoplasia of treated larvae with POP125×; (**D**) dose-based prevalence of eye malformation; (**E**) prevalence of eye malformation in larvae upon treatment with the different POP sub-mixtures. Data is presented as mean percentage of population having eye malformation and standard deviation; ordinary one-way ANOVA and Šidák’s multiple comparison test, *n* = 100, *p* ≤ 0.001 (***). Asterisks indicate when significant differences were found compared to control, hash sign (#) when differences were found compared to PFAA. (**F**) Prevalence of eye malformation present in fish exposed to POP125× relative to untreated individuals. Columns represent the effect of POP125 treatment in larvae previously injected with Mo^*p53*^ (POP125× + injection) or not (POP125×). A Fisher’s exact test revealed the difference with a significance of *p* = 0.055 (*).

**Figure 4 toxics-11-00357-f004:**
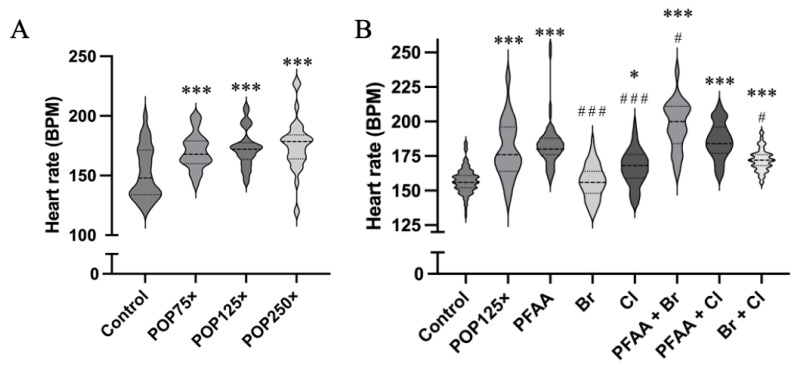
(**A**) Heart rate (BPM, beats per minute) of treated and untreated zebrafish using 3 different concentrations of the total POP mix. (**B**) Heart rate of all the fish exposed to the different treatments and measured at 4 dpf. Kruskal–Wallis test and Dunn’s multiple comparison, *n* = 30, *p* < 0.05 (*), *p* ≤ 0.001 (***). Asterisks indicate when significant differences were found compared to control, hash (#) sign when differences were found compared to PFAA.

**Figure 5 toxics-11-00357-f005:**
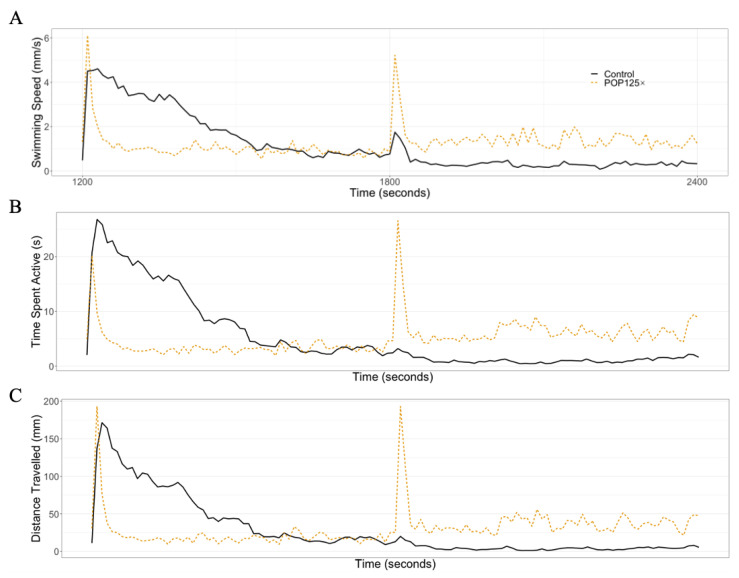
Actogram example of (**A**) swimming speed, (**B**) time spent active, and (**C**) distance travelled during 20 min of the tests, starting with 10 min dark phase followed by 10 min light. For comparison, control larvae are shown alongside POP125×-treated larvae. Two spikes can be appreciated when the fish entered a different phase.

**Figure 6 toxics-11-00357-f006:**
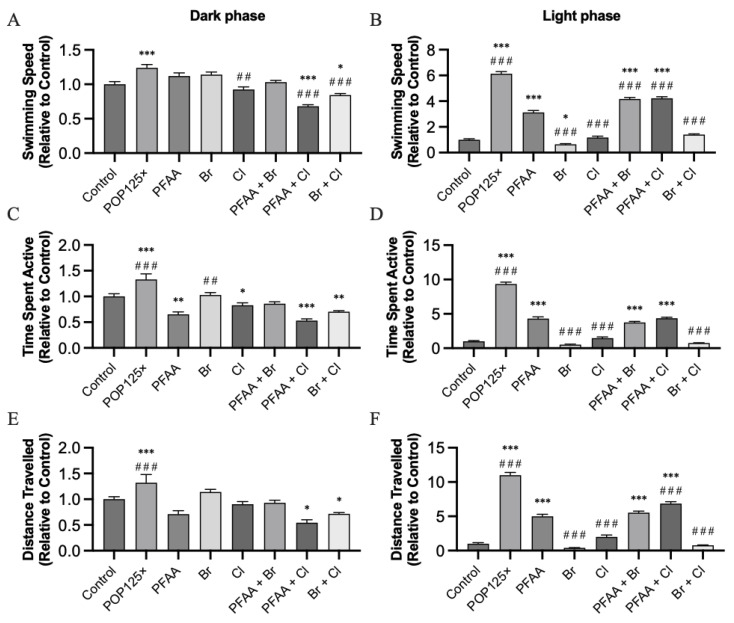
Behavior tests during the dark (**left**) and light (**right**) phases: swimming speed (**A**,**B**), time spent active (**C**,**D**), and distance travelled (**E**,**F**) for all treatments. All data were calculated excluding 50 s in the transition zone between light and dark phases, and the values were normalized relative to the corresponding controls for dark and light phases, *n* = 72, *p* < 0.05 (*), ≤ 0.01 (**), ≤ 0.001 (***). Asterisks (*) indicate when significant differences were found compared to control, hash (#) sign when differences were found against PFAA.

**Figure 7 toxics-11-00357-f007:**
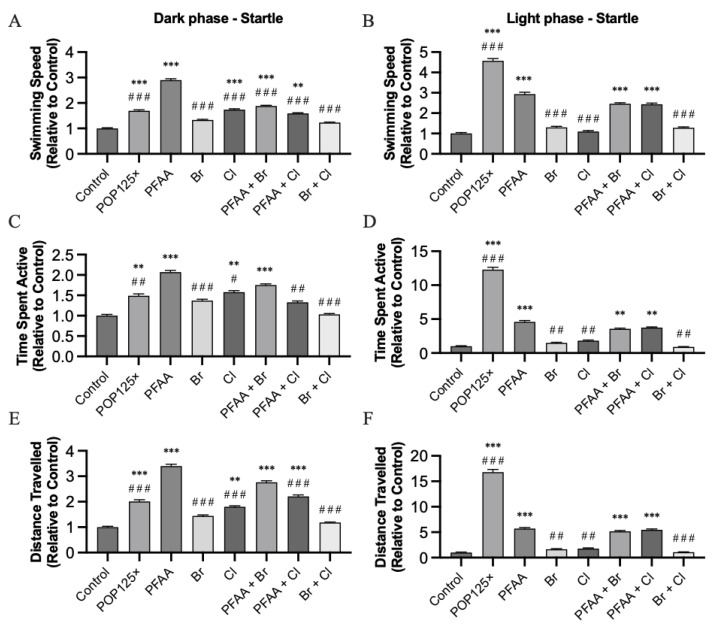
Swimming speed (**A**,**B**), time spent active (**C**,**D**), and distance traveled (**E**,**F**) for all treatments. Left column, parameters during the dark phase, right column, parameters during the light phase. All results were calculated using 50 s during the transition zone between light and dark phases, *n* = 72, *p* ≤ 0.01 (**), ≤0.001 (***). Asterisks (*) indicate when significant differences were found compared to control, hash sign (#) when differences were found against PFAA.

**Figure 8 toxics-11-00357-f008:**
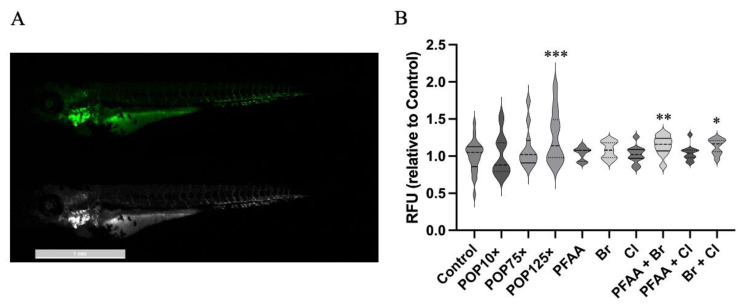
(**A**) Transgenic *Tg(kdrl-mls:GFP)* fish line treated with POP125×. (**B**) Plot with median values of the normalized fluorescence intensity of fish treated with different POP mixes at 96 hpf. There is a clear increasing trend. Kruskal–Wallis test and Dunn’s multiple comparison, *n* = 21, *p* < 0.05 (*), *p* < 0.01 (**), *p* < 0.001 (***).

**Figure 9 toxics-11-00357-f009:**
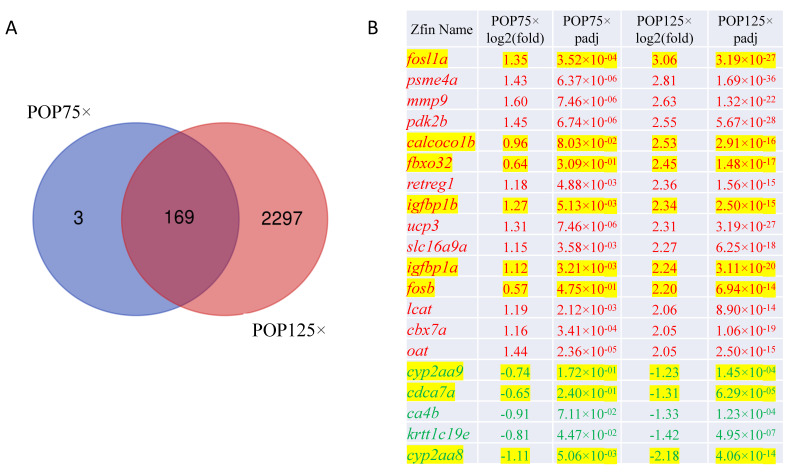
(**A**) Venn diagram comparing the lists of genes affected by POP1 or POP2 treatment. (**B**) Most highly and significantly regulated genes upon treatment with both POP total mix concentrations at 5 dpf. Log(fold change) and significance (adjusted *p*-value) are shown. Upregulated genes are in red, downregulated in green, while genes discussed in the text are highlighted in yellow.

**Figure 10 toxics-11-00357-f010:**
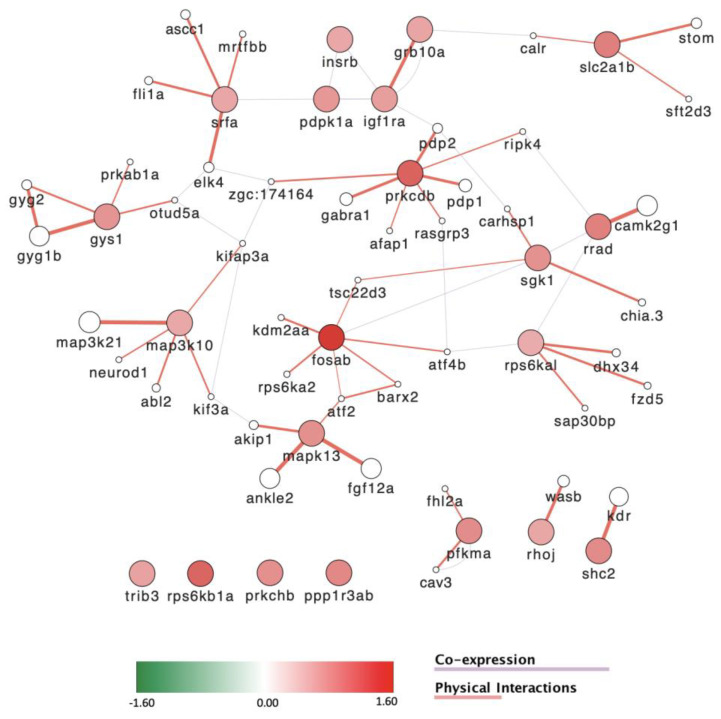
Differentially expressed genes involved in insulin signaling and response. These genes are all upregulated upon treatment with POP125× and distribute into specific co-expression and physically interacting modules.

**Figure 11 toxics-11-00357-f011:**
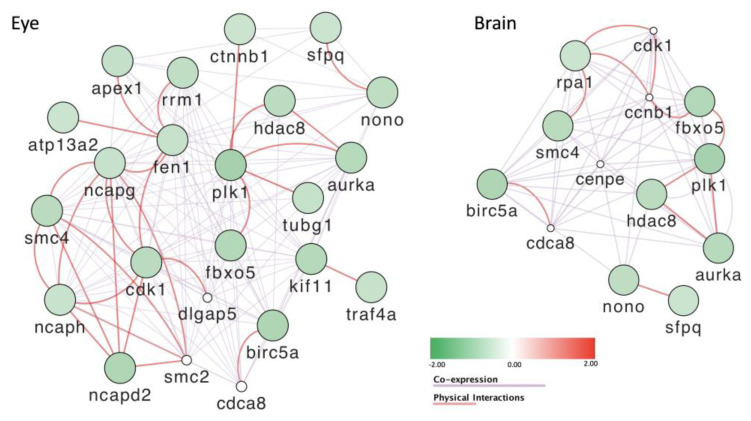
Differentially expressed genes that are involved in brain and eye development. These genes are all downregulated in zebrafish larvae treated with POP125× and form co-expression and physically interacting networks.

## Data Availability

Raw and processed sequencing data have been deposited in NCBI’s Gene Expression Omnibus [92] and are accessible through GEO Series accession number GSE208019 (https://www.ncbi.nlm.nih.gov/geo/query/acc.cgi?acc=GSE208019; accessed on 8 April 2023).

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
