# Peer review of "A Realistic Mixture of Persistent Organic Pollutants Affects Zebrafish Development, Behavior, and Specifically Eye Formation by Inhibiting the Condensin I Complex"

_toxics, 2023, doi:10.3390/toxics11040357_

Round 1
Reviewer 1 Report
This manuscript is an answer to often suggested investigations. It offers an insight in the toxicity of combinations of compounds. Therefore, it is very useful.
The investigation is soundly designed and performed. The results are presented clearly and the conclusion is correct and very useful.
This manuscript needs no further comments.
Author Response
Dear reviewer 1
Thanks for this positive evaluation. We really appreciate.
No action needed here
Reviewer 2 Report
The methods should be improved. The concentrations used of the mixture should be well described. The informations related to the treatment as described for exaple in lines 205-210 of the results sections, sould be discussed in the methods section
